# Improving screening recall services for women with false-positive mammograms: a comparison of qualitative evidence with UK guidelines

Mary Bond,[1] Ruth Garside,[2] Christopher Hyde[1]

[1]University of Exeter Medical School, University of Exeter, Exeter, UK
[2]University of Exeter Medical School, University of Exeter, Truro, UK

**Correspondence to**
Mary Bond;
m.bond@exeter.ac.uk

## ABSTRACT

**Objectives:** To gain an understanding of the views of women with false-positive screening mammograms of screening recall services, their ideas for service improvements and how these compare with current UK guidelines.

**Methods:** Inductive qualitative content analysis of semistructured interviews of 21 women who had false-positive screening mammograms. These were then compared with UK National Health Service (NHS) guidelines.

**Results:** Participants' concerns about mammography screening recall services focused on issues of communication and choice. Many of the issues raised indicated that the 1998 NHS Breast Screening Programme guidelines on improving the quality of written information sent to women who are recalled, had not been fully implemented. This included being told a clear reason for recall, who may attend with them, the length of appointment, who they will see and what tests will be carried out. Additionally women voiced a need for: reassurance that a swift appointment did not imply they had cancer; choice about invasive assessment or watchful waiting; the offer of a follow-up mammogram for those uncertain about the validity of their all-clear and an extension of the role of the clinical nurse specialist, outlined in the 2012 NHS Breast Screening Programme (NHSBSP) guidelines, to include availability at the clinic after the all-clear for women with false-positive mammograms.

**Conclusions:** It is time the NHSBSP 1998 recall information guidelines were fully implemented. Additionally, the further suggestions from this research, including extending the role of the clinical nurses from the 2012 NHSBSP guidelines, should be considered. These actions have the potential to reduce the anxiety of being recalled.

## INTRODUCTION

Screening for breast cancer by mammography has been part of many women's routine healthcare for more than 25 years. Much research has been carried out into the anxiety produced by having a false-positive mammogram (FPM).[1–6] However, the quality of mammography screening services for women who have been recalled has been less thoroughly investigated. Internationally, questionnaire studies have found that overall women are satisfied with the service they receive.[7–10] Additionally, some studies found that the attitude of clinic staff as well as the quality of information and the physical environment affected satisfaction.[7] [9–12] Furthermore, a Finnish observational study that investigated the information needs of women assessed by biopsy, found that women wanted information and reassurance throughout and after their assessment.[11] However, a Canadian randomised controlled trial failed to find an impact on satisfaction from additional information.[13] The above research gives an opaque picture of the information and support needs of women recalled following screening.

The situation in the UK is particularly unclear as our searches found only one UK study of service satisfaction of women with a FPM. This was by Smith *et al*[14] who found that clinic staffs' attitudes, quality of information and the physical environment had an impact on satisfaction. However, this survey is

### Strengths and limitations of this study

- This research has been rigorously conducted by an independent, academic research team.
- The suggestions for service improvements are based on empirical research.
- The evidence provides current insights into women's view of mammography screening services for recalled women.
- The study may be limited by the ability of some participants to recall distant experiences.
- More detailed information about the demographic characteristics of participants would aid interpretation of the results.

more than 20 years old and it is 16 years since the NHS Breast Screening Programme (NHSBSP) produced guidelines about the information needs of women recalled following mammography screening.[15] More recently NHSBSP guidelines (2012) highlight the important role of the clinical nurse specialist (CNS) in supporting women who have been recalled.[16] As FPM affects more than 50 000 women a year in England alone, we were interested in women's views of the service they received, their thoughts on how they might be improved and how these compared with NHSBSP guidelines.

The research question is: What are the views of women with false-positive screening mammograms of the recall service they received, their ideas for service improvements and how do these compare with existing UK guidelines?

## METHOD

We chose to use semistructured interviews because they employ open-ended questions within the framework of an interview guide, facilitating a discourse where the interviewee is free to respond to the questions in a self-directed way. This approach produces responses that are rich in content and may contain interesting and relevant material beyond the scope of the initial question.

### Participants and recruitment

Fifty-two women with FPM were invited to participate and 21 were recruited (40%). Recruitment was through the National Institute for Health Research Primary Care Research Network, from three local general practices or through the University of Exeter staff e-newsletter. Participants were purposively sampled for diversity of age, time from the false-positive experience and type of assessment procedure. We were also interested in the social mix of participants and used the UK Index of Multiple Deprivation (IMD), relating to their post code, as a means of assessing this. The IMD is derived from a national survey of income, employment, health, education, housing, crime and living environment. The scores are ranked from the least to the most deprived.[17] Owing to the specific focus of the research it was believed that about 20 interviews would be sufficient for data saturation. If saturation did not occur further participants would be recruited.[18] Participants gave informed consent.

### Data collection

Participants were interviewed by MB in quiet locations of their choosing, usually at home. The interview guide (available from the authors) was used to gather key pieces of information. It covered the experiences of being invited for screening, being recalled, the assessment clinic and reflections of that experience. The guide was based on the results of the latest UK systematic review[1] and reviewed by two women with FPM. The interviews were recorded and transcribed.

### Data analysis

The interviews were analysed with inductive qualitative content analysis.[19] This approach was chosen because we wanted to develop simple categories from the interviews to compare with the items in the guidelines rather than explore the deeper meanings of what the participants were saying. This process involved reading and listening to the interviews iteratively as relevant content was open coded. The codes were reviewed across the manuscripts by a process of constant comparison, being merged and dropped as the analysis progressed. The codes were then gathered into categories of similar items.[20] These primary categories were subsumed into higher order generic categories and so assisted the systematic description of the phenomena, thus identifying the key messages in the texts.[21]

The results were validated using Yardley's principles of sensitivity to context, commitment and rigour, transparency and coherence and impact and importance, including an audit trail and the search for disconfirming cases[22] and participant feedback. The analysis was supported by Atlas.ti V.6.2 software. The application of Yardley's principles can be seen in table 1.

## RESULTS

Participants' characteristics were found to fulfil the criteria of diversity. However, more detailed information about educational level, income and social group would aid interpretation of the results, see table 2.

The interview study showed that overall; almost all participants were satisfied with the mammography recall service they received. However, as they presented their stories a number of issues were raised for service improvement. These issues concerned the recall letter, the assessment clinic, choice and subsequent screening. Participants' quotes are identified by a pseudonym. A diagram of the relationship between the categories can be found in figure 1.

### Recall letter information

Most women were satisfied with the quality of the recall information they were sent. The information was repeatedly described as reassuring. Many participants latched onto the positive messages of the letter and remembered they had been told that most recalled women were clear of breast cancer and they should not worry about being called back.

> Anne: The letter itself I think said something reassuring like… 'as a precaution we're calling you back because, there's some anomaly, or something like that, on the screen…' and then it says … a large proportion of women who are called for second screening don't actually have anything, but it's just a precautionary thing, so

**Table 1** Yardley's principles for quality in qualitative research

| Principle | Qualities | Application to this study |
|---|---|---|
| Sensitivity to context | Theoretical; relevant literature; empirical data; sociocultural setting; participant's perspective; ethical issues | Grounding the study in the context of what is already known from the systematic review. Then gathering the new interview data to refine that knowledge, searching for examples that confirm and refute what is already known. Being aware of and sensitive to the sociocultural place of the participants and how this might influence the meanings they give to their experiences. Also understanding those experiences from their perspective; what they meant to them, but acknowledging the influence of the researcher and their role in the interview to what is said through their demeanour, verbal and non-verbal cues and an awareness of the potentially more powerful position of the researcher. Ethical approval was gained |
| Commitment and rigour | In-depth engagement with topic; methodological competence; skill; thorough data collection; depth/breadth of analysis | This is achieved through becoming committed to the process of the research, the integrity of the interviews, being emersed in the data and taking a systematic, rigorous approach to the depth of analysis and interpretation of the interviews. Accounting for the variety and complexity of the data, including the search for disconfirming cases. The trustworthiness of the analysis was further established by respondent validation of the findings. The first eight interviews were coded independently by two researchers |
| Transparency and coherence | Clarity and power of description/argument; transparent methods and data presentation; fit between theory and method; reflexivity | Providing a clear audit trail of the process of the study including data analysis. Telling a clear coherent story that encompasses the range of experience of the participants, illustrated by their own words and offering a reflective interpretation of the meaning of their accounts that acknowledges the role and influences of the researcher |
| Impact and importance | Theoretical (enriching understanding); sociocultural; practical (for community, policymakers, health workers) | Clearly describing the originality and importance of the findings and how they relate to previous research. Demonstrating their importance for policymakers, in this case the NHSBSP. Offering recommendations to improve services to reduce the psychological impact of false-positive mammograms |

Source: Yardley (2000).

I mean all the time they're kind of trying to put forward the, sort of, positive angle on it.

However, there was considerable variation in what women were told to expect; some were simply told they would have another mammogram and others were given an explanation about the reason for their recall.

Clare: In the letter, yeah, when I was recalled, they actually said … 'we'd like you to come again, so we can take some further x-rays,' and they've also said 'at this visit we may also carry out an ultrasound examination', so I was aware of what was going to happen.

Moira: Only curious about what was going to happen, because you don't get told beforehand. You know you're going to have a mammogram, but you don't know what else is going to happen… so you're not actually prepared.

Most of the women found the short time between the recall letter and their assessment a positive thing as the time of anxiety was curtailed:

Vicky: When I got the letter, to be recalled was only a few days later, it wasn't very long, which I'm really glad about. I couldn't have hung on two or three, four weeks, it was a matter of days, it was very, very good, very good.

Conversely, others interpreted the quick appointment as possibly indicating they had breast cancer. This understanding was rooted in a belief that the NHS only responds quickly to serious health problems:

Grace: There wasn't a lot of time, um, also between the letter—I think it was about a week—between me getting the letter and actually going for the, the next mammogram, which tends to sort of go 'oops,' anything that comes you don't have much time in between means that it could be serious.

Others who went alone found the waiting hardest as they lacked the support of a friend or relative, but not everyone knew they could bring someone with them.

Wendy: The only thing that I would have liked the letter to have said was, 'if you want to bring a friend or relative, please do so.

These accounts indicate a variation in the amount and content of the information that participants were given. They also show that the information needs of these women differed; some were able to focus on the positive messages of the recall letter and were quite happy with simply being told that although they were being recalled

**Table 2** Summary of participants' characteristics

| Characteristics | Women, N (%) |
|---|---|
| Age (years) | |
| 40–49 | 2 (10) |
| 50–59 | 11 (52) |
| 60–69 | 8 (38) |
| Marital status | |
| Married or cohabiting | 19 (90) |
| Single, separated or widowed | 2 (10) |
| Ethnicity | |
| White | 21 (100) |
| Time since false-positive (years) | |
| ≤1 year | 4 (19) |
| 2–4 | 7 (33) |
| 5–7 | 8 (38) |
| 8–10 | 1 (5) |
| 11–13 | 1 (5) |
| Type of assessment procedure* | |
| Mammogram | 16 (76) |
| Ultrasound | 12 (57) |
| Fine needle aspiration | 1 (5) |
| Biopsy | 4 (19) |
| Index of Multiple Deprivation % | |
| Unknown | 4 (19) |
| 0–9 | 1 (5) |
| 10–19 | 0 (0) |
| 20–29 | 2 (9) |
| 30–39 | 3 (14) |
| 40–49 | 2 (9) |
| 50–59 | 4 (19) |
| 60–69 | 4 (19) |
| 70–79 | 1 (5) |

*Many women had more than one assessment procedure.

everything was probably alright. Others reacted more strongly to the uncertainty that was introduced into their lives and wanted as much detail as possible about the reasons for their recall; the implications of this and what was going to happen at the assessment clinic.

### At the clinic

Waiting generally provoked anxiety; one participant said this could have been reduced by information about how long the clinic appointment was likely to take:

Karen: The waiting was the worst…the whole thing was bad, but the worst bit was having to wait and not knowing how long I was waiting for.

Many participants reported that the clinic staff were the best thing about their experience. They were repeatedly described as Rachel: lovely, Moira: friendly, Ella: supportive, Zoe: kind, Anne: professional and Vicky: very, very nice:

Zoe: They were just very kind and I think, in a way, nurturing, because they knew there was a possibility that you might have bad news. They were just very protective of you; you just felt that they were handling it really well, that they cared about you.

Clear explanations of the nature of the lesion were valued and helped to bring peace amid the uncertainty.

Laura: He was ever so…really thorough, I've got to say, really put my mind at rest … explained everything to me from start to finish, … and did it in a way…not condescending way, he explained it in a real clear and concise manner, yeah, absolutely brilliant. I came out cock-a-hoop!

Although the majority had positive encounters, not all the staff were sensitive and caring.

Chloe: Um…yes, it didn't feel quite friendly, the staff in the unit sometimes…it was a case of, 'oh, yeah,' it's just… like the queue… 'oh, right, yeah, next.

There was evidence of an unmet need for information and reassurance that could have been given by a CNS being available, and known to be available, after as well as before assessment. This unavailability resulted in some women leaving the clinic with uncertainty and unanswered questions.

Laura: Don't think so [someone to talk to]. Didn't see… there was a nurse, yes, there was a nurse on duty, but she was very busy and I could see she was very busy.

Grace: I can't recall there being anybody around.

Fran: Well, I suppose somebody to talk…you know, perhaps if I'd gone and talked to somebody, you know, about it all, perhaps I…it would have eased me.

During their assessment the amount of information that participants wanted about their lesion varied. A number of them would have had more confidence in their all-clear result if they had been given a clearer explanation of what their lesion meant, including the risk of it becoming malignant so they could make an informed choice about how to proceed.

Liz: I would like the doctors to present me with the facts and say 'right, you know…if it's 20 women out of…out of 100 with hyperplasia who, you know, before they die, so they could be really old, get breast cancer,' then I know that fact. If it's, um, you know, one in 10,000 up to the age of 80, then I probably would opt not to have any invasive surgery again.

One participant's experience highlighted the need for consistency between the messages from the clinical staff and the literature they were given. The radiologist had told her that her cysts could not become cancerous and then gave her a leaflet that said they could; this caused considerable anxiety. This inconsistency led her to request an interim mammogram for reassurance from her general practitioner (GP) but was turned down:

Moira: And I now wait for my next mammogram…but that's the bit I don't like. You're told you have cysts, but

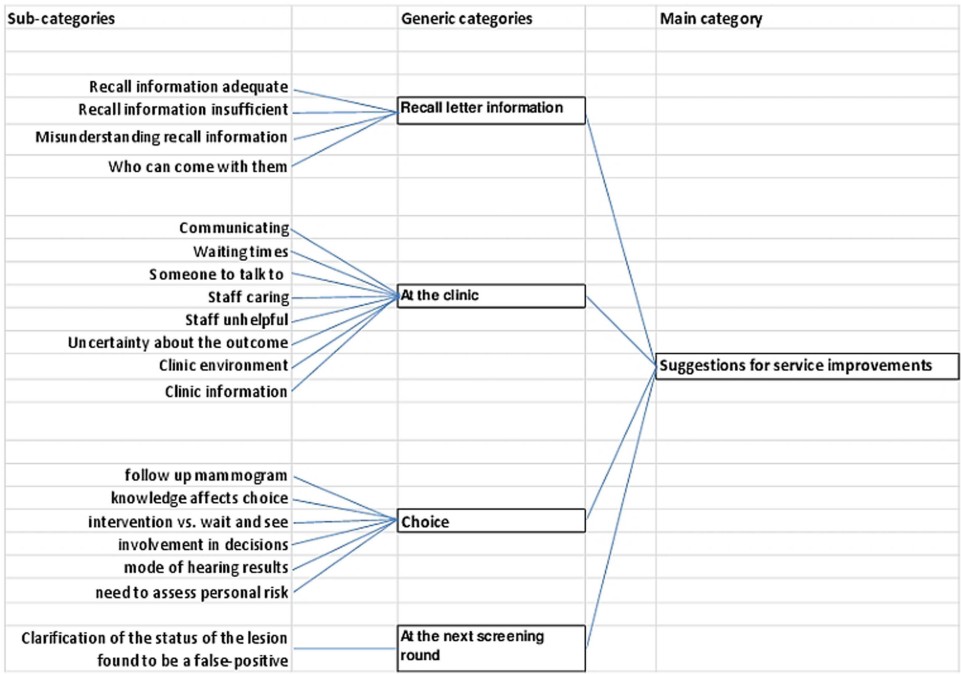

**Figure 1** Category development.

you're not given a follow-up from there and you should…I'm assuming that they're quite satisfied that they [cysts] cannot become cancerous, even though the leaflet says they can. So I'm assuming they either haven't read the leaflet, or they simply don't believe the leaflet, I don't know which.

While the women waited at the clinic it was apparent that the location, layout and the waiting environment played a role in moderating their experience. Some participants had their initial screening in mobile units but then went to a hospital for their assessment.

Zoe: Perhaps it seems more serious because you're aware it's a place where there are sick people… whereas if you go to the unit that goes to the car parks and whatever, you're in and out in ten minutes and you move on.

For some the physical layout of the clinic meant they were aware of what was happening to other women ahead of them in the queue; increasing their anxiety as they waited.

Grace: We waited in a corridor … we, sort of, waited in a long line, sat down in a long line … we all waited together and we all went in and you'd see people go through the door, but they also came out that door and you knew if it was good or bad news for them, because obviously if it was good news they were looking happy, and if it was bad news they didn't go straight down the corridor, they went into another room. Um, and that would have been, I think, pretty traumatic for anybody, because you knew which way…but also for us, because we knew if it was good or bad news for them. I remember

thinking, 'well, I'm going to have to go through that in a minute, but I don't want to witness their distress either.

On other occasions the participants could not see the woman diagnosed with breast cancer but they could hear her crying, which similarly increased their anxiety.

Wendy: while I was waiting somebody else had…who had been recalled, um, had been told that they, you know, they did have something and obviously they were distressed and, you know, people waiting, that was distressing as well…I could hear, I couldn't see her, but I could hear her and that was distressing, yes.

Although the staff received much praise, there was still a clear need for an opportunity to talk to the CNS after the all-clear if questions remained about the reliability of the diagnosis and the probability of it becoming malignant. The status of their lesion remained a concern at the next screening round for some women and information about if or how it had changed would have been valued. The responses also showed that a sensitivity to the clinical environment is necessary with due regard for privacy.

### Choice

Following their assessment a number of women were uncertain about the validity of their all-clear diagnosis. They would have liked the option of a follow-up mammogram for reassurance.

Ella: I thought 'can you just see me in a year's time, just tell me that in a year's time, it's all OK in a year's time?

Zoe: I think the fact that you've been faced with the possibility that something didn't look quite right, you're not quite sure what it was that didn't look quite right, um, and maybe a screening a year down the line would have been, um, something to…to, you know, relieve any nerves.

There was also a request for mammograms as an alternative to the invasive investigation of lesions by biopsy that were thought to be almost certainly benign.

Liz: Because they said, 'we're pretty sure it isn't anything, but we need to check if it is anything.' …and so I suppose what I'm saying is that had there been a little bit more, um, wait and see… I wouldn't have had more anxiety if they'd say, 'well, we'll do a mammogram again in three months or six months.'

This plea for choice was part of a desire to be more involved in the decision-making process about how to proceed once a lesion was detected. Many of the women felt disempowered by the assessment process. They perceived themselves to be Ella: 'in the system' and passive partakers of assessment procedures.

Liz: I think when you're in the middle of it, you just go along with whatever's being told… there could have been, uh, more consultation maybe at the beginning of things… so…and I'd have probably still have gone along with it, [surgery] 'cause I don't think I felt empowered not to.

Another woman would have appreciated the choice of receiving her biopsy results by post or over the phone, as this further wait prolonged anxiety.

Fran: It was just waiting for those results… every day you look for the post… and when it's there you're frightened to open it up…it [phone call] probably would have been better, really, 'cause you're looking every day, aren't you, at the post and thinking, 'oh, my gosh.'

It was clear that some women would have valued more choice in the assessment process. Choice was requested for follow-up mammograms to reduce uncertainty and anxiety about the outcome of assessment, 'watchful waiting' as an alternative to biopsy and test results over the phone rather than through the post.

### At the next screening round

At their next routine screening, some participants' anxieties would have been dispelled if they were told about the status of the lesions previously discovered:

Jane: I just got the all-clear letter, sort of thing, [after subsequent screening] and when I saw the doctor [GP] I said 'when I…you got the results of me mammogram did it say anything about the cyst?' And he looked it back and he said, 'no, it hasn't said anything.' And it would have been just nice to know whether I've still got it or not.

### Comparison of these results with NHSBSP guidelines

Many of our findings echo those of the research used to produce the NHSBSP 1998 guidelines and indicate that these have not been universally implemented.[23–25] A comparison of the NHSBSP guidelines with our service improvement suggestions can be found in table 3.

The following suggestions are offered to improve services.

### Recall letter

Recall letters issued should be consistent and include the following items. The first five items remain unaddressed from the 1998 guidelines.

This should include

1. The reason for recall.
2. Who can come with them.
3. How long the appointment is likely to take.
4. Who they will see.
5. What tests will be carried out.
6. Where to get further information.
7. The availability of a CNS to answer questions before and after assessment.
8. Reassurance that a swift appointment is normal and does not indicate there is anything wrong.

### At the clinic

9. A preassessment conversation with the CNS covering, the reason for recall, the assessment process including possible harms, and the availability of the CNS for a debrief after the assessment whatever the outcome.
10. From the Radiologist at diagnosis; sufficient time for a clear explanation of the type of lesion, risk of it becoming malignant, with clarity about uncertainty. If a biopsy is advised then discussion about pros and cons including the reliability of biopsy results and the choice of watchful waiting if the lesion is almost certainly benign.
11. The availability of CNS post-assessment to clarify the diagnosis and provide reassurance, as a woman may not feel able to question the outcome with her GP. If the woman remains unsure of the validity of her all-clear a follow-up mammogram should be considered.
12. Literature about the type of lesion found should be offered. This should agree with that from the Radiologist and give a phone number for further information.
13. The choice of receiving biopsy results by phone or post.
14. A one-way system through the clinic so that women do not have to have the outcome of their assessment witnessed or witness other's outcomes.

### At the next screening round

15. Women should be given an update about their lesion, whether it has gone, stayed the same or grown larger, with an explanation of the implications.

**Table 3** Comparison of our suggestions with those of the 1998 and 2012 NHSBSP guidelines

| | NHSBSP Recall guidelines (38) 1998 | Bond *et al* service suggestions 2014 |
|---|---|---|
| Recall letter information | A clear reason for recall | A clear reason for recall |
| | Who can come with them | Who can come with them |
| | How long the appointment will take | How long the appointment will take |
| | Who they will see | Who they will see |
| | What tests will be carried out | What tests will be carried out |
| | Where they can get further information | Reassurance that a swift appointment does not imply the presence of cancer |
| | How to get to the assessment centre | |
| | How to change their appointment | |
| | When the results will be available | |
| At the clinic | NHSBSP CNS guidelines (29) 2012 | |
| | Availability of a CNS before assessment | Availability of a CNS before and after assessment |
| | | Clear explanation of why the lesion is benign with any risk of change to malignancy |
| | | Literature about the type of lesion |
| | | One-way layout through the clinic |
| Choice | | The offer of a follow-up mammogram in a year for those needing reassurance of their 'all clear' |
| | | Choice between invasive assessment and 'watchful waiting' for lesions almost certainly benign |
| | | Choice of hearing biopsy results by post or over the phone |
| At the next screening round | | Clarification of the status of the lesion found to be a false-positive |

NHSBSP, National Health Service Breast Screening Programme.
Grey shading represents the items from the 1998 NHSBSP Guidelines that were found to be still outstanding in the interviews, ie the same matters arose.

## DISCUSSION

While overall participants were satisfied with the service they received, they raised a number of areas where it could be improved. Many of the items in the 1998 NHSBSP guidelines remained outstanding. Some women were still asking for more information in the recall letter about: the reason for their recall; who could come with them; how long the appointment might take; who they would see and what would happen to them. Some participants were also concerned that a swift appointment implied that they had cancer. At the assessment clinic a more explicit explanation of why their lesion was benign and the risk of it becoming malignant would have reduced anxiety, together with literature about their type of lesion. The women also expressed requests for more choice as some were left with doubt about the validity of the outcome of the assessment and would have appreciated an offer to have a follow-up mammogram in a year's time for reassurance and an update on the status of their lesion at their next screening round. Others felt powerless and in 'the system' when faced with a biopsy; an alternative of 'watchful waiting' for lesions that were almost certainly benign, would have been valued and empowering. A choice of having biopsy results by post or over the phone was also requested. Additionally some participants would have valued an opportunity to see the CNS after assessment as well as before. Finally, the layout of the clinic precipitated anxiety for some women while they were waiting, as they were able to see and hear the distress of women who had been diagnosed with cancer. There was no apparent link between participants' demographic characteristics and the issues they raised.

The strengths of this study are that it was rigorously conducted and provides current, in-depth insights into the views of women, with FPMs of the assessment service they received. Thus it provides valuable evidence of how these services may be improved so that the anxiety associated with having FPMs may be reduced. The influence of the researcher (MB) is acknowledged, both from her manner, verbal and non-verbal cues during the interview; there is also the potential for social desirability effects.[26] The analysis has been through the filter of the researcher's particular understanding of the issues, including being someone

who has not had mammography. This will have influenced the way the interviews were interpreted, due to the lack of first-hand experience. Qualitative research is notably subjective and is open to the charge that the results lack generalisability. However, although the study has a number of limitations and further research is needed to establish the UK national picture, we believe these results are reasonably robust, transferable and relevant for consideration in policy development. This is because the interview findings were validated using Yardley's principles[22] and by participant feedback of a lay summary of the results. The study is limited by lack of demographic detail, the limited geographical area (Devon) and the possible unreliability of the participants' memories as these events occurred between 6 months and 12 years previously that is, recall bias.[27] However, the consistency of our findings with previous research[23–25] and other studies, which have shown a positive association between the accuracy of long-term recall and the traumatic impact of an event up to 21 years,[28 29] give us confidence in the reliability of our results.

Internationally, questionnaire studies have also found that overall women are satisfied with mammography screening recall services.[7–10] In the case of FPMs there is a certain irony about this response as the women are satisfied with a service that has made a mistake in recalling them and may have caused them unnecessary anxiety. Further evidence comes from the Danish interview study by Lindberg *et al*[30] which found women with FPMs were grateful for the service which had brought their health into question and caused them psychological distress. Some studies found, in agreement with ours, that the attitude of clinic staff as well as the quality of information and the physical environment affected satisfaction.[7 9 10 14] Our findings also agree with the results from the US qualitative study (2001), that some participants thought the information they received was inadequate.[12]

Although our research comes from a limited geographical area, and other regions of the UK may have better service provision, it implies that there is still some way to go to provide women who are recalled after breast cancer screening with a satisfactory service. There is a need for consistency in the implementation of recommendations and a mechanism for ensuring this occurs.

## Service implications

These suggestions will require additional resources; increased hours for the CNS and additional mammography for those needing reassurance or choosing 'watchful waiting'. However, most of the suggestions can be implemented at the lower cost of revising literature and giving clearer explanations, which may mean that fewer women are left with uncertainty and request on-going care.

## CONCLUSION

It is time the NHSBSP 1998 recall information guidelines were fully implemented. Additionally, the further

suggestions from this research, including extending the role of the CNS from the 2012 NHSBSP guidelines, should be considered.

Further research is needed to establish whether the 1998 NHSBSP recommendations are in place nationally and if the additional measures outlined in these service recommendations are sought by women throughout the UK.

**Contributors** MB obtained ethical approval, obtained NIHR adoption, recruited participants with the support of the SW Primary Care Research Network, conducted and analysed the interviews, obtained respondent feedback, undertook the comparison with UK guidelines and formulated service improvement suggestions, wrote and edited the research paper. RG contributed to the design of the study, advised on the qualitative aspects and commented on the draft paper. CH contributed to the design of the study, advised on the health services aspects and commented on the draft paper.

**Funding** This paper presents independent research funded by the National Institute for Health Research (NIHR). This was partially through their Collaboration for Leadership in Applied Health Research and Care (CLAHRC) for the South West Peninsula and partially through the HTA Programme.

**Competing interests** None.

**Ethics approval** This study received ethical approval from the UK National Research Ethics Service Committee South West, approval no. 11/SW/0263.

**Provenance and peer review** Not commissioned; externally peer reviewed.

**Data sharing statement** No additional data are available.

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
