## [Reviewer comments · BMJ Open]

Some articles will have been accepted based in part or entirely on reviews undertaken for other BMJ Group journals. These will be reproduced where possible.

ARTICLE DETAILS

TITLE (PROVISIONAL)	Improving screening recall services for women with false-positive mammograms: a comparison of qualitative evidence with UK guidelines
AUTHORS	Bond, Mary; Garside, Ruth; Hyde, Chris

VERSION 1 - REVIEW

REVIEWER	John Brodersen The Research Unit and Section of General Practice Department of Public Health Faculty of Health Sciences University of Copenhagen Øster Farimagsgade 5 P. O. Box 2099 DK-1014 Copenhagen Denmark
REVIEW RETURNED	14-Aug-2014

GENERAL COMMENTS	Page 2, abstract, conclusion, line 42-49 It is time the NHSBSP 1998 recall information guidelines were fully implemented. Additionally, the further suggestions from this research, including extending the role of the clinical nurses from the 2012 NHSBSP guidelines, should be considered. These actions have the potential to reduce the anxiety of being recalled. Re: this conclusion does not describe the actual results from the present study and what conclusion(s) can be drawn for these results. The content of the present conclusion in the abstract covers two areas: "implication for practice" and "implication for research". Page 3, line 15-16: The study is limited by the small sample size: Re: A sample size of 20 is not a small sample size in a qualitative study. It is not the numbers of informants in a qualitative study that is a limitation it is the variance among informants and if data saturation is reached. In the present study there are three serious limitations: all informants come from the same geographical area, a lot of them may suffer from recall bias and the variance is not good enough among the informants since many of them may come from the same socio-demographic level.
--

Page 4, line 14-22:

It is hypothesised that the role that adequate information and personal support play in increasing service satisfaction, is due to their ability to reduce uncertainty and thus anxiety. For people with acute and chronic illness, uncertainty is acknowledged to be the greatest cause of stress.^{17;18} It is likely that this is also the case with recall following screening. As Warren says, 'a woman who receives a recall letter experiences temporarily the diagnosis of cancer'.

Re: this background information does not seem to have any relation with the actual research question and the hypothesis raised (although it is very relevant and might be true) here cannot be answered and confirmed by the presents study's research question and design. Therefore, I would recommend the authors to either skip this topic, or move it to the discussion paragraph and maybe even write a paragraph called "implication for research" where the authors could suggest a randomized controlled trial to be conducted confirming or not confirming the hypothesis raised.

Page 4, line 48-52:

Participants were purposively sampled for diversity of age, time from the false-positive experience and type of assessment procedure.

Re: please justify why: "time from the false-positive experience" was included as a criterion? Firstly, if it many months or some years ago the women had the false-positive experience there is a great chance of recall bias. The authors actually also state this in the discussion: "the possible unreliability of the participants' memories" (page 19, line 2-4). Secondly, the research question is: "What are the views of women with false-positive screening mammograms of the recall service they received, their ideas for service improvements and how do these compare with existing UK guidelines?". If a purpose is to compare the women's views with the existing guidelines then I would certainly prefer that only women that had recently had a false-positive screening mammography were interviewed. Because, what if the services had improved during the last year and you had interviewed women that had their experiences longer than a year ago? Would you then still recommend the services to be changed?

Another issue that strikes me here is why socio-demographics are not reported especially education level, income or social group. There is robust evidence showing that there is a great inequality in healthcare services and therefore I would guess that women from different social groups would experience the services in relation to downstream procedures following a false-positive screening mammography to be very different. Why did you not stratify your informants according to the socio-demographic and if you have the

information about the women's socio-demographics please add this information to Table 2.

Please enter these limitations to the manuscript, e.g., at page 3, line 17-18 in the sentence about limited geographical location.

Page 6, line 39-40

Participants' characteristics were found to fulfil the criteria of diversity

Re: See my comments about this issue elsewhere.

Page 17 and 18, line 43 – 25

Re: this first paragraph in the discussion is more or less just a resume of the findings of the study with no discussion of these findings. For example it is mentioned "The women also expressed requests for more choice as some were left with doubt about the validity of the outcome of the assessment and would have appreciated an offer to have a follow-up mammogram in a year's time for reassurance and an update on the status of their lesion at their next screening round." The scientific value of this wish of an annual follow-up mammography could easily have been discussed according to the present best available evidence. Therefore, the paragraph should either be substantially shorten and only encompass the main results from the present study describe in not more than 7-8 lines – or all the results should be relevantly discussed according to best available evidence.

Page 18, line 43-49

However, we believe these results are robust, transferable and relevant to policy development as the interview findings were validated using Yardley's principles²⁴ and by participant feedback on a lay summary of the results.

Re: I do disagree with this quite strong statement from the author. I think my previous comment about the recall bias and the lack of information about the socio-demographics should result in a more humble way of describing the validity of the present study's findings.

Page 19, line 6-8

Internationally, questionnaire studies have also found that overall women are satisfied with mammography screening recall services.

Re: what is missing here is the dependency and the asymmetric relationship there is between the women with abnormal screening mammography results and the staff at the recall clinics. This paradox: that women with false-positive screening mammography express a great thankfulness to a preventive initiative and a medical technology that has made a mistake (a false-positive result) and the great relief the women also have expressed in several studies after

	being declared free from cancer suspicion (the women’s own verbal expression) has to be included in the present study’s discussion about “satisfaction”. Is it actually satisfaction that has been measured in the referenced studies? This topic should be well-known to the first author of the present study because she touches the topic of measurement in her systematic review (reference 1 in the present study). Therefore, it is also quite disappointing that nor my own longitudinal 3-year follow-up survey about long-term psychosocial consequences neither my colleagues and my qualitative study interviewing women 4-5 years after the false-positive screening results are included in the present paper background description and/or discussion. These two papers stand on several solid qualitative and psychometric previous published studies where the issue about validity of measurement has been thoroughly discussed and to a certain extent solved in my longitudinal 3-year follow-up survey including more than 1,300 women. Page 19, line 40-52, Conclusion Re: Please read my previous comment to the conclusion in the abstract.
--	--

REVIEWER	Katriina Whitaker University College London, UK
REVIEW RETURNED	07-Oct-2014

GENERAL COMMENTS	Thank you for the opportunity to review this manuscript exploring the views of women with false positive screening mammograms of screening recall services. With very high rates of FPM in the national screening programme, there is no doubt this is an important issue. Unfortunately I have some reservations about the manuscript that I will outline below. Major concerns The aim of this study was to learn more about the views of women using services following false positive mammograms to inform the development of national guidelines. However, the sample size is small, (n=21) and is very homogenous (e.g. 70% married, 100% White ethnic origin), so it is unlikely the results can have the impact on policy that are claimed (e.g. extending role of nurses). I’m not convinced that the study design is appropriate to answer the research question- perhaps survey methodology would have helped capture the experiences of a more representative proportion of the 50,000 women in England experiencing FPM each year (e.g. satisfaction of information provision, attitudes to clinic visit, confidence in their ‘all clear’). Although including a qualitative component would certainly be an important adjunct to this approach in order to capture women’s experiences in more depth. My second major concern is the time lapse between women’s experiences of services and their involvement in the study. If FPM is so common, why were such a significant proportion of the sample more than 1 year post their FPM (and several more than 5 years)? Perhaps it
--

	would have been more informative to talk to people currently using services to avoid problems with recall/retrospective bias. Other concerns  • The results sound quite quantitative, e.g. 'all women' satisfied (p7), 'Most' women were satisfied' (p8), which supports the view that survey methodology may have been more suitable. • How was purposive sampling used, and why wasn't there purposive sampling for other demographics? The age range doesn't seem particularly diverse and the sample is very homogenous, limiting generalisability. • Index of Multiple Deprivation needs more explanation and discussion in the methods. • Results, p6 "Participant's characteristics were found to fulfil the criteria of diversity, see table 2". What criteria? Based on what guidelines? • Discussion, p18: "There was no apparent link between participants demographic characteristics and the issues they raised" How was this tested? I'm not sure this can be asserted from the results. • Discussion, p18 "These results are robust, transferable and relevant to policy development." Unfortunately, the study design does not fit with the claim that these findings are transferable.
--	---

VERSION 1 – AUTHOR RESPONSE

Reviewer 1

1 Page 2, abstract, conclusion, line 42-49.

The reviewer correctly says that the abstract's conclusions do not describe the results. These are described in the results section of the abstract and the discussion section of the manuscript. The conclusion in the abstract reflects that of the manuscript i.e. that changes to practice need to be considered. It is usual for the conclusions to cover implications for practice and research.

2 Page 3, line 15-16: The study is limited by the small sample size:

We agree with the reviewer that 20 is not a small sample size in a qualitative study and have removed this statement.

3 All informants comes from the same geographical area:

It is quite common in qualitative research for participants to come from the same geographical area, we disagree that this is necessarily a limitation. Further, we would like to emphasize that this is the first qualitative study of the general mammography screening population in the UK. Therefore it was reasonable to take this initial exploratory step in our local region, before going further afield.

4 A lot of them may suffer from recall bias:

We agree that this is a consideration which we acknowledged as a limitation in the discussion and have also included in the summary of strengths and limitations section. This will always be the case whether an interview or questionnaire study is undertaken, particularly with long-term follow-up, which are dependent on people's memories. However, the recalled memory is important as this is how respondents currently view their experience and will thus impact on current and future behaviour in relation to screening. Moreover, to rule out studies that rely on memory would be to effectively exclude any study based on interviews or questionnaires. In this case most of the

participants had their false-positive mammogram within the last four years. We also note that this reviewer is an author of a Danish interview study of eight women with false-positive mammograms who were interviewed five years after the event (Lindberg et al. 2013). Additionally, previous research has shown that duration of effect is an important consideration (Bond et al. 2013); therefore we considered it important not to limit the length of time from the event when recruiting.

5 The variance may be not good enough among the informants since many of them may come from the same socio-demographic level:

The participants were purposively sampled to ensure diversity. Table 2 (p6) has information about the participants' characteristics which shows good variation in the sample including that they were from a wide range of socio-economic backgrounds. The exception to this is ethnicity. However, as this is the first interview study in this population in the UK, we believed (as stated above) that it was reasonable to sample from women in our local region. In Devon the population is almost entirely white (93%). We did not exclude people on the basis of race; however, only white women came forward. Moreover, an underlying purpose of this research was to see if there were issues that needed further exploration on a wider scale. As we said in our conclusion, further research is needed to see if the views we found locally are reflected nationally.

6 Page 4, line 14-22: This background information does not seem to have any relation with the actual research question and the hypothesis raised (although it is very relevant and might be true) here cannot be answered and confirmed by the presents study's research question and design:

We concede that while this paragraph is relevant to the overall research field it does not have a direct bearing on the research question and have removed it.

7 Page 4, line 48-52: Please justify why "time from the false-positive experience" was included as a criterion?:

The point made by the reviewer about current time since false positives is relevant; although we can reassure that the screening programme in the UK has remained relatively static over many years. The reason we included time from false-positive experience as part of our sampling frame was because we wanted to see if there was any influence of time over the kinds of issues that women raised and explore the duration of effect; in the event time from the false-positive did not seem to affect the issues raised. Again we would emphasise that this study is a first step in largely uncharted waters in the UK, so we were throwing the net wide to see what was out there. As noted above our conclusions speak of the need for further confirmatory (or otherwise) research. Please see our comments above about the impact of memory on recall.

8 Another issue that strikes me here is why socio-demographics are not reported especially education level, income or social group:

All the demographic information that we collected has been shown in Table 2. We agree that educational level would have been an interesting addition. Although the index of multiple deprivation scores may be a proxy for this, income and social group. However, we do not consider that stratification is appropriate in a qualitative study.

We had added a statement 'more detailed information about the demographic characteristics of participants would aid interpretation of the results' to the summary strengths and limitation section on p 3. And a further statement on page 6: 'However, more detailed information about educational

level, income and social group would aid interpretation of the results’.

9 Page 17 and 18, line 43 – 25:

It is standard practice to summarise the results of a study in the opening paragraph of a discussion section.

We agree that it is important to establish whether the views the participants disclosed in these interview have scientific merit; this would make an interesting paper. However, the purpose of this paper is to compare the views of women with false-positive mammograms of the services they received and their ideas for service improvements with the UK guidelines. As this is a qualitative paper much of this discussion is found in the results section.

10 Page 18, line 43-49; I do disagree with this quite strong statement from the author:

We have acknowledged that the study has limitations and agree to extend this acknowledgment by revising this statement thus:

‘However, although the study has a number of limitations and further research is needed to establish if the results reflect the national picture, we believe they are reasonably robust, transferable and relevant for consideration in policy development’.

11 Page 19, line 6-8:

We are grateful to the reviewer for pointing out this omission. We have corrected this by adding the following to the discussion section:

‘In the case of false-positive mammograms there is a certain irony about this response as the women are satisfied with a service that has made a mistake in recalling them and may have caused them unnecessary anxiety. Further evidence comes from the Danish interview study by Lindberg et al. which found women with false-positive mammograms were grateful for the service which had brought their health into question and caused them psychological distress.’

We are aware of the reviewer’s major contribution to the literature on measuring psychological outcomes in women with false-positive mammograms but are unclear of its place with reference to this qualitative study.

12 Page 19, line 40-52, Conclusion. Re: Please read my previous comment to the conclusion in the abstract:

Please see our response to this comment above

Reviewer 2

13 As this is a qualitative interview study 21 is not a small sample size. This number of interviews was sufficient to achieve data saturation, if it had not been we would have carried on recruiting. We agree with the reviewer that all of the sample were white and most married or cohabiting. We believe this is defensible as this is the first qualitative study of this population in the UK. It is therefore reasonable to sample from the local population. In the case of Devon this is 93% white. We have acknowledged that the views we found may not reflect the national picture and stated that further research is needed. This research is an initial step in qualitative research in the UK.

14 We agree that survey methods have an important role to play in understanding women’s views. Indeed this method was used in the 1990’s to gather the information that the 1998 NHSBSP

guidelines were based on. The limitation of this method is that you only get answers to the questions you ask. Therefore, qualitative research has a different role to play and is able to uncover concerns that may be overlooked in a questionnaire. As it is more than 15 years since this survey work was done we consider that a local interview study is a reasonable approach to explore women's views of the services they received.

15 We agree that this is a serious consideration which we acknowledged as a limitation in the discussion and have also included in the summary of strengths and limitations section. This will always be the case whether an interview or questionnaire study is undertaken, particularly with long-term follow-up, which are dependent on people's memories. However to rule out studies that rely on memory would be to effectively exclude any study based on interviews or questionnaires. In this case most of the participants had their false-positive mammogram within the last four years. We sampled for a wide range of time from experience partly to see if there was any indication of an effect of time on the kinds of things the participants were saying.

16 We disagree with this comment, that 'most' and 'all' are only quantitative terms and that this is evidence for a questionnaire study. Indeed, quantitative research would require numbers rather than such terms. It is common in reporting qualitative research to give some indication of how consistently themes are found across the analysed interviews.

The purposive sampling was used to select a range of women who had false-positive screening mammograms. We were particularly interested in diversity of age, time from event and assessment procedure as we considered these would enable a wider range of experience than simply sampling anyone with this experience. We did not exclude on any grounds beyond the essential of having a false-positive screening mammogram. The lack of ethnic diversity reflects the Devon population, although we could have sampled for a greater mix of single and married women. We disagree that the sample is very homogenous, the age range corresponds to that of the screened population, there is a range of assessment procedures undergone, which has a direct bearing on the service received and the sample were from a wide social mix as evidenced by the Index of Multiple Deprivation scores. We have not claimed this sample reflects the national demography and have said in the conclusion that further research is needed to see if our results are found nationally.

17 We are happy to expand on this and have added the following in the methods:

'We were also interested in the social mix of participants and used the UK Index of Multiple Deprivation (IMD), relating to their post code, as a means of assessing this. The IMD is derived from a national survey of income, employment, health, education, housing, crime and living environment. The scores are ranked from the least to the most deprived'.

18 This refers to our criteria for diversity i.e. a range of ages, methods of assessment and time from event.

19 This was assessed by comparing the kinds of issues raised with the participants' characteristics. Please note that we said there is no 'apparent' link, we are not claiming that there is no link, just that we were not able to detect one.

20 We have acknowledged that this study has limitations and have expanded on this, see above.

However, we hold that the results are transferable to similar women in similar situations. None-the-less this needs to be tested, as we have said, in different locations. This is not the same as saying the results are generalisable to all women.

VERSION 2 – REVIEW

REVIEWER	John Brodersen The Research Unit and Section of General Practice Department of Public Health Faculty of Health Sciences University of Copenhagen Denmark
REVIEW RETURNED	24-Nov-2014

GENERAL COMMENTS	I do not think the authors have changed their manuscript enough according to the commennts made by me and the other review. Especially the issue about recall bias is totally misinterpreted by the authors. They state: "We agree that this is a consideration which we acknowledged as a limitation in the discussion and have also included in the summary of strengths and limitations section. This will always be the case whether an interview or questionnaire study is undertaken, particularly with long-term follow-up, which are dependent on people’s memories. However, the recalled memory is important as this is how respondents currently view their experience and will thus impact on current and future behaviour in relation to screening. Moreover, to rule out studies that rely on memory would be to effectively exclude any study based on interviews or questionnaires. In this case most of the participants had their false-positive mammogram within the last four years. We also note that this reviewer is an author of a Danish interview study of eight women with false-positive mammograms who were interviewed five years after the event (Lindberg et al. 2013). Additionally, previous research has shown that duration of effect is an important consideration (Bond et al. 2013); therefore we considered it important not to limit the length of time from the event when recruiting." Our reserach question in the paper by Lingberg et al. was about how the women's experiences were AT THE MOMENT 4 to 5 years after a false positve screening mammography. We DID NOT ask them had the remembered their experiences 4 to 5 years ago. This is was is done in the present paper. Therefore, recall bias was NOT an issue in our paper but is in the present paper. And a recall period of in average 4 years is too fare away.
--